# Frequent miRNA-convergent fusion gene events in breast cancer

Helena Persson [1], Rolf Søkilde [1], Jari Häkkinen [1], Anna Chiara Pirona[1,4], Johan Vallon-Christersson [1], Anders Kvist [1], Fredrik Mertens [2], Åke Borg[1,3], Felix Mitelman[2], Mattias Höglund[1,3] & Carlos Rovira[1,3]

Studies of fusion genes have mainly focused on the formation of fusions that result in the production of hybrid proteins or, alternatively, on promoter-switching events that put a gene under the control of aberrant signals. However, gene fusions may also disrupt the transcriptional control of genes that are encoded in introns downstream of the breakpoint. By ignoring structural constraints of the transcribed fusions, we highlight the importance of a largely unexplored function of fusion genes. Here, we show, using breast cancer as an example, that miRNA host genes are specifically enriched in fusion genes and that many different, low-frequency, 5′ partners may deregulate the same miRNA irrespective of the coding potential of the fusion transcript. These results indicate that the concept of recurrence, defined by the rate of functionally important aberrations, needs to be revised to encompass convergent fusions that affect a miRNA independently of transcript structure and protein-coding potential.

[1] Department of Clinical Sciences Lund, Oncology and Pathology, Lund University Cancer Center, Lund University, SE-223 63 Lund, Sweden. [2] Department of Clinical Genetics, Skåne University Hospital, Lund University, SE-221 85 Lund, Sweden. [3] BioCARE, Strategic Cancer Research Program, SE-223 63 Lund, Sweden. [4] Present address: German Cancer Research Center, Division of Functional Genome Analysis, Heidelberg, Germany. Correspondence and requests for materials should be addressed to C.R. (email: carlos.rovira@med.lu.se)

Chromosomal aberrations resulting in the fusion of two different genes are common somatic alterations in cancer[1]. They were first discovered in hematologic malignancies, but have also been identified as important drivers in sarcomas as well as in tumors of epithelial origins. The typical outcome of fusions genes is the merging of coding parts of two different genes to produce a chimeric protein with new properties. However, examples of fusions that only exchange regulatory regions are also known (for a review see ref. [2]). The study of gene fusions began more than 30 years ago, when the Philadelphia chromosome in chronic myeloid leukemia (CML) was shown to result in a BCR–ABL1 chimera and the MYC gene was found to be deregulated through fusions with immunoglobulin-encoding genes in lymphomas[3, 4]. Since then, technical advances have fueled the search for fusion genes in cancer and aided in their identification. Today, the use of next-generation sequencing has resulted in a formidable explosion in the number of characterized gene fusions[5] and a range of bioinformatic tools have been developed for fusion gene discovery in RNA-seq data[6]. Most algorithms for detection of fusion genes are specifically designed to study annotated genes and chimeric transcripts that retain protein-coding capacity[7]. Meanwhile, fusions that involve unannotated partners or form out-of-frame fusion transcripts are often discarded from further analysis. This approach will exclude genetic components where the transcription and processing from the primary transcript rather than protein-coding potential are the essential functional components. One example is the microRNA (miRNA) class of non-coding RNAs, which have been shown to promote carcinogenesis[8–14]. Notably, almost 60% of human miRNA genes are encoded within introns and 84% of these are influenced by the same regulatory DNA elements as the coding part of the gene[15]. Precursor miRNAs (pre-miRNAs) are processed co-transcriptionally from their host transcript[16–18]. Therefore, gene fusions that place any miRNA downstream of the breakpoint will put it under the control of the 5′ partner gene promoter regardless of the protein-coding potential and position of the breakpoint in the fusion transcript.

To the best of our knowledge, the effects of gene fusions on miRNA regulation have been largely disregarded. We therefore performed a genome-wide study to analyze to what extent genomic fusion transcripts also include miRNA genes, using breast cancer as a model. Our results show that genes hosting miRNAs are overrepresented among aberrant fusion transcripts and that 802 pre-miRNA loci in 667 host genes were part of expressed fusion transcripts in breast cancer. We also show that fusions are associated with aberrant expression of the miRNA. Some of these fusions have previously been described as recurrent protein-coding chimeras, while the inclusion of the intronic miRNAs has been ignored. Importantly, many of the fusions hosting intronic miRNAs might have been disregarded because they are non-recurrent in a strict sense. Here we show that even if individual fusions including miRNA host genes may occur at low frequencies, they are in fact recurrent in the sense of being "miRNA convergent". Although the exact fusion partners and breakpoints vary, they have in common that they can deregulate the expression of the same miRNA. In summary, our results add a new level of complexity to our understanding of the functional consequences of fusion genes in malignancies.

## Results

**Identification of miRNAs in fusion transcripts.** We used FusionCatcher[19] to identify candidate fusion transcripts in a set of 1552 breast tumors sequenced by strand-specific mRNA-seq within the Sweden Cancerome Analysis Network—Breast[20] (SCAN-B)[20]. In total, we identified more than 400,000 fusion transcripts whereof 8% were in-frame fusions, 9% out-of-frame fusions, and 53% included protein-coding DNA sequences (CDS) predicted to encode truncated proteins. The remaining fusions involved different combinations of RNAs transcribed from intergenic regions, introns, untranslated regions (UTRs) or regions of unknown coding potential according to FusionCatcher classification (see Supplementary Table 1 for details). In total, 11,424 unique genes were involved in fusion transcripts in at least one tumor (Supplementary Table 2).

The full set of 1881 pre-miRNA loci from miRBase v21[21] was mapped to host genes using GENCODE v22[22] annotation before comparison between their genomic coordinates and all identified fusion transcript breakpoints. A total of 1275 primary miRNAs (68%) could be assigned to 1104 host genes, and an additional 45 (2%) were located within 2 kb downstream of a candidate host gene. As many as 802 primary miRNAs (61% of miRNAs mapped to a host gene) were included in fusion transcripts in at least one tumor. Applying a more stringent criterion, 514 (39%) were found in fusions with maintained relative 5′ or 3′ position and recurrent in at least three tumors. The number of miRNAs involved in fusion genes is summarized in Table 1 and a full list of these miRNAs, their host genes, and fusion partners is included in Supplementary Data 1. Note that the breakpoints used here represent the exon–exon junctions detected by mRNA-seq and not the true genomic breakpoints. Precursor miRNAs that are located in the intronic sequence between a 5′-to-3′ fusion partner splice junction will therefore be missed by our analysis, even if transcribed as part of the fusion.

False positives are common in fusion transcript prediction and FusionCatcher includes a number of filtering steps to remove these (see "Methods" for details). The software also flags fusion genes previously identified in normal tissue samples as "healthy" and excludes most of these from further analysis. Our final

## Table 1 MicroRNA precursors encoded in host genes with fusion transcripts in our data and in TCGA data

|  | Any fusion | 5′ Fusion partner | 3′ Fusion partner |
|---|---|---|---|
| ≥1 Tumor | 802 (61%) | 691 (52%) | 640 (48%) |
| ≥1 Tumor and in TCGA | 163 (12%) | 155 (12%) | 153 (12%) |
| ≥1 Tumor and in TCGA breast | 142 (11%) | 66 (5%) | 80 (6%) |
| Recurrent (≥3 tumors) | 514 (39%) | 426 (32%) | 380 (29%) |
| Recurrent and in TCGA | 124 (9%) | 119 (9%) | 92 (7%) |
| Recurrent and in TCGA breast | 101 (8%) | 51 (4%) | 47 (4%) |
| Recurrent our data and TCGA | 44 (3%) | 24 (2%) | 18 (1%) |
| Recurrent our data and TCGA breast | 14 (1%) | 8 (1%) | 6 (0.5%) |

Number of miRNA precursors encoded in host genes with fusion transcripts with the corresponding percentage of all 1320 analyzed pre-miRNAs in parenthesis. 5′ or 3′ fusion partner refers to the position of the miRNA host gene and the column "any fusion" combines all precursors present in 5′ and/or 3′ fusion partner genes. Recurrent fusions were defined as occurring in at least three tumors with the host gene in the same position

filtered set contained 250 fusions flagged as "healthy" (0.06%) involving the six genes *CDH11*, *COL1A1*, *FGFR1*, *FN1*, *NCOR2*, and *UBC*. Of these six genes, only *NCOR2* is a miRNA host gene and encodes *mir-6880*, but only one out of 82 samples with fusions involving *NCOR2/mir-6880* had transcripts that had also been found in healthy tissues.

We also validated our set of miRNA host gene fusions by comparison with independent cancer data and analyzed a published set of fusion transcripts available in The Cancer Genome Atlas (TCGA) Fusion Gene Data Portal. A total of 359 (32%) and 66 (6%) miRNAs out of 1108 with an assigned host gene were found in fusions in at least one or three tumors of any cancer type, respectively. When the comparison was restricted to breast cancer, 171 (15%) and 20 (2%) miRNAs were found in fusions in at least one or three tumors, respectively. The numbers of miRNAs in fusion genes that have supporting evidence in this database are included in Table 1 and information about the fusion transcripts is included in Supplementary Data 2.

The overlap between our data and all cancer types in TCGA was highly significant, both for miRNAs within a fusion transcript in at least one tumor (odds ratio 2.93, 95% CI [2.15–4.03], $p = 3.29 \times 10^{-13}$, Fisher's exact test), for miRNAs recurrent in our data and with at least one fusion in TCGA (odds ratio 2.99, 95% CI [2.29–3.93], $p < 2.20 \times 10^{-16}$, Fisher's exact test), as well as for miRNAs recurrent in both data sets (odds ratio 3.03, 95% CI [1.75–5.39], $p = 2.62 \times 10^{-5}$, Fisher's exact test). It remained significant also when reducing the analyzed set of miRNAs to a single precursor per host gene in order to avoid the potential bias caused by clustering of miRNAs within genes; the corresponding odds ratios for miRNAs within a fusion transcript in at least one tumor was 2.59 (95% CI [1.87–3.62], $p = 1.62 \times 10^{-9}$, Fisher's exact test), for miRNAs recurrent in our data and with at least one fusion in TCGA 2.73 (95% CI [2.06–3.62], $p = 5.46 \times 10^{-13}$, Fisher's exact test), and for miRNAs recurrent in both data sets 2.61 (95% CI [1.48–4.71], $p = 0.00042$, Fisher's exact test). Many of the miRNA host genes that were involved in fusion transcripts in our breast tumor data were also found in other cancer types such as lung and prostate adenocarcinoma in the TCGA data (Supplementary Data 2).

The overlap was still significant when the comparison was limited to breast cancer samples in the TCGA data, both for miRNAs within a fusion transcript in at least one tumor (odds ratio 2.77, 95% CI [1.80–4.39], $p = 3.79 \times 10^{-7}$, Fisher's exact test), for miRNAs recurrent in our data and with at least one fusion in TCGA (odds ratio 2.34, 95% CI [1.66–3.32], $p = 5.01 \times 10^{-7}$, Fisher's exact test), as well as for miRNAs recurrent in both data sets (odds ratio 3.38, 95% CI [1.21–10.82], $p = 0.01$, Fisher's exact test). It remained significant also when the set of miRNAs was reduced to a single precursor per host gene; the corresponding odds ratio for miRNAs within a fusion transcript in at least one tumor was 2.75 (95% CI [1.75–4.47], $p = 1.81 \times 10^{-6}$, Fisher's exact test) and 2.22 for miRNAs recurrent in our data and with at least one fusion in TCGA (95% CI [1.55–3.18], $p = 4.90 \times 10^{-6}$, Fisher's exact test). Only 12 miRNAs with unique host genes were recurrent in both data sets and this overlap was not significant (odds ratio 2.82, 95% CI [0.97–9.22], $p = 0.05$, Fisher's exact test). These analyses confirm that a substantial fraction of miRNA genes are frequently part of gene fusions in cancer.

**MicroRNA host genes are overrepresented among fusion genes.** We used logistic regression to model the relationship between the probability of a gene being involved in a fusion transcript and its status as a miRNA host gene. To minimize bias, the analysis was limited to genes expressed in our breast tumor set and annotated

by GENCODE as protein coding. Gene size was also included in the model. The size-adjusted odds ratios for miRNA host genes vs non-host genes to be involved in gene fusions were 2.29 for all fusions (95% CI [1.93, 2.70], $p < 2.00 \times 10^{-16}$, Wald test) and 2.48 for fusions recurrent in at least three samples (95% CI [2.16, 2.86], $p < 2.00 \times 10^{-16}$, Wald test). However, since longer genes are also more likely to encompass miRNAs[23, 24] (Supplementary Fig. 1), we proceeded by modeling fusion probability as a function of miRNA host gene status, gene size and the interaction between the two. All three factors were highly significant, indicating that gene fusion probability differs between miRNA host genes and non-host genes, but that the effect varies with gene size. As shown in Fig. 1a, b, miRNA host genes were associated with a higher gene fusion probability for gene sizes below ~440 or 730 kb, respectively, when considering all fusions or recurrent fusions. This includes 91% and 96% of all miRNA host genes or 91% and 95% of all miRNAs included in the analysis, respectively. By contrast, logistic regression failed to show any significant association between gene fusions and the presence of several other genetic elements that occur with a similar frequency within protein-coding genes according to RepeatMasker annotations (Fig. 1c–h). In summary, these results indicate that miRNA host genes are indeed overrepresented among genes involved in fusion transcripts.

**Host gene enrichment is not associated with protein function.** In-frame fusions are biased towards certain types of genes[25, 26]. We therefore investigated if the observed enrichment of miRNA host genes in fusion transcripts was associated with the functional characteristics of the encoded proteins, rather than the presence of the miRNAs. The set of host genes that contained a miRNA and specifically identified as 3′ fusion partner in at least three tumors within the data set (328 genes with an Entrez gene identifier) was compared with a background set consisting of all genes with fusions in at least three tumors as either 5′ or 3′ partner (6391 genes with an Entrez gene identifier) by Gene Set Enrichment Analysis[27] (GSEA) using clusterProfiler[28]. No significant or very marginal enrichment was found for the proteins encoded by these miRNA host genes for Gene Ontology terms (cellular component, molecular function, and biological process), Disease Ontology terms (DOSE), the Kyoto Encyclopedia of Genes and Genomes (KEGG), the Reactome Pathway Database, or the Molecular Signatures Database (MSigDB) (Supplementary Data 3). These results suggest that it is the included miRNAs rather than the proteins encoded by their host genes that are the functional elements selected for in tumor cells.

**Fewer miRNAs within 3′ partner genes of in-frame fusions.** Intronic miRNAs may regulate their host genes through different feedback mechanisms[23]. This would also be true for in-frame fusion transcripts with a functional miRNA in the 3′ partner. Our data show that in-frame fusion transcripts are less frequent among fusion genes that include a miRNA within the 3′ partner compared to fusions where the 3′ partner gene does not contain a miRNA. By contrast, there was no difference between 5′ fusion partners that did or did not contain miRNAs within the fused gene segments (Fig. 2a). Since chimeric transcripts have also been found in normal cells we wanted to know whether this was a particular characteristic of cancer fusions and analyzed a set of fusion transcripts discovered in normal samples from ~30 different tissue types[29]. These showed no depletion of in-frame fusions for cases where the 3′ partner gene includes a miRNA (Fig. 2b). As shown in Fig. 2c, the fraction of genes with in-frame fusions that also contain miRNAs is 9% lower than among genes with any type of fusion (5.6 vs 6.1%) in our breast cancer data.

The difference increased to 20% when only comparing the miRNA content in 3′ fusion partners (4.5 vs 5.6%). The percentage of miRNA-containing fusions was considerably lower in the normal tissue data and there was no difference between in-frame fusion transcripts and all fusions for genes with any type of fusion (both 4.3%) and only small differences for 5′ (4.3 vs 4.4%) and 3′ (3.8 vs 3.9%) fusion partners. Taken together, these data indicate that preservation of a reading frame selects against the inclusion of miRNAs in fusion genes and this specifically affects the fusions that might drive miRNA expression.

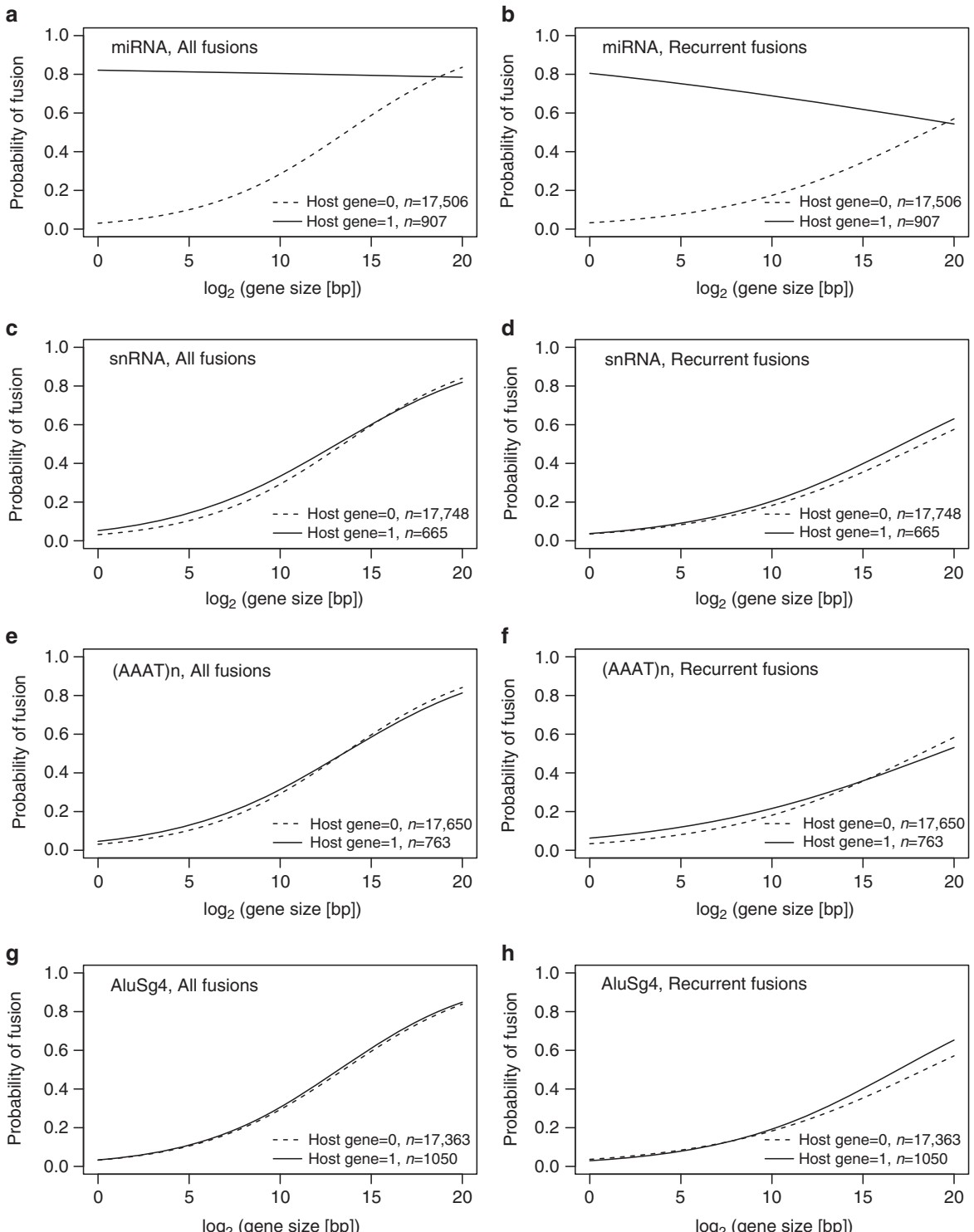

**Fig. 1** MicroRNA host genes are overrepresented among fusion genes. Logistic regression with a model including host gene status, gene size, and the interaction between the two, showed that miRNA host genes were significantly more likely to be involved in fusion transcripts, both when considering all fusions (**a**) or recurrent fusions (**b**). No significant association was found between gene fusions and the presence of several other genetic elements that occur with a similar frequency in protein-coding genes according to RepeatMasker annotation (**c**–**h**)

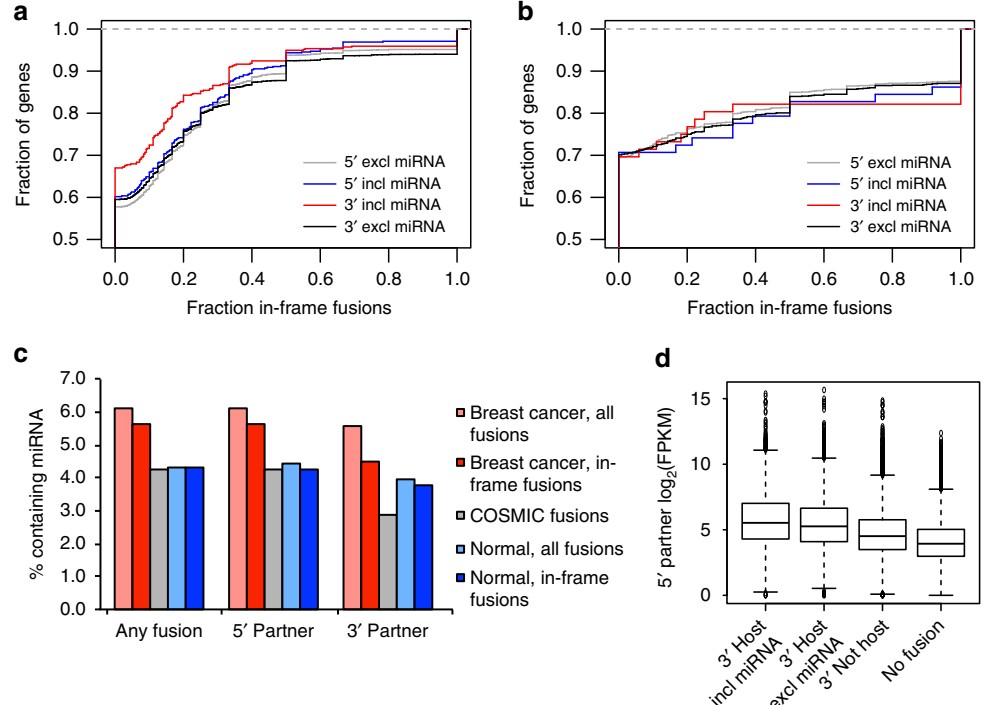

**Fig. 2** Fusion transcripts involving miRNA host genes as 3′ partners have fewer in-frame fusions. Cumulative distribution of the fraction of in-frame fusions among all fusion transcripts, plotted separately for 5′ and 3′ partner genes that lack or include miRNAs within the fused gene segments in our breast cancer data (**a**) and among fusion transcripts from approximately 30 different normal tissue types from ref. [27] (**b**) Percentage of genes with fusions that include miRNAs when analyzing all fusion types or only in-frame fusion transcripts for all protein-coding genes involved in fusions or among 5′ and 3′ partners separately, and, for comparison, among manually curated fusions in the COSMIC database (**c**). The 5′ partners of miRNA host genes have higher average expression than those of non-host genes, especially when the miRNA is included in the fused segment of the 3′ partner (**d**). Average expression was calculated separately for each 5′ partner gene among tumors with a fusion transcript matching either of the three 3′ partner categories. No fusion represents expression for the full set of 5′ partner genes in samples without fusion transcripts involving them. Expression is in $\log_2$(FPKM) (fragments per kilobase of exon model and million reads)

We also compared the expression of 5′ partners for three different fusion gene categories based on the properties of the 3′ partner: (1) host genes with the miRNA included within the fused gene segment, (2) host genes lacking the miRNA within the fused segment, and (3) non-host genes (Fig. 2c). The expression of 5′ partners was significantly higher for 3′ host genes including the miRNA than for 3′ host genes excluding the miRNA ($4.66 \times 10^{-6}$, Student's $t$-test), or for expression of 3′ non-host genes ($p < 2.2 \times 10^{-16}$, Student's $t$-test).

**Underrepresentation of miRNA host genes in fusion databases**. Our findings suggest that a focus on identification of chimeric proteins from in-frame fusion transcripts will introduce a bias against fusions that could deregulate miRNA expression. To assess this effect, we analyzed fusion events from publicly available databases. We first queried the Mitelman Database of Chromosome Aberrations and Gene Fusions in Cancer[30] for all breast cancer fusion genes and found that 244 genes out of 2580 5′ fusion partners (9%) and 237 genes out of 2959 3′ fusion partners (8%) contained miRNAs. Results were similar when we restricted the analysis to fusion genes that have been described in at least two independent reports; we found nine miRNA host genes among 98 5′ fusion partners (9%) and six among 94 3′ fusion partners (6%). The overlap with the fusions identified in our set of breast tumors is shown in Table 2 and the corresponding miRNAs are marked in Supplementary Data 1. Without information on the genomic coordinates for breakpoints it was not possible to calculate how many of these fusion genes that

actually include the miRNA. These figures may therefore slightly overestimate the actual number of fusions in the database that include miRNAs.

The FusionCancer database[31], however, contains predicted fusion transcripts for 594 samples from 15 different cancer types with publicly available RNA-sequencing data. The results have not been experimentally validated but include genomic coordinates for the identified fusion breakpoints. We found that 39 out of 1082 5′ fusion partners (4%) and 44 out of 1071 3′ partners (5%) included miRNAs within the predicted fused gene segments. The corresponding numbers for gene fusions recurrent in at least three samples of any cancer type were 15 out of 333 (5%) for 5′ fusion partners and 13 out of 303 (4%) for 3′ fusion partners. Precursor miRNAs found in fusion transcripts are marked in Supplementary Data 1.

Lastly, we also analyzed fusions reported in the manually curated COSMIC database[32], which included 10,435 fusion events in its latest release. As seen in Fig. 2b, COSMIC consistently contained fewer fusions that contained miRNAs than our data, irrespective of the position of the miRNA in relation to the fusion breakpoint. In agreement with the results reported above, the difference is more pronounced when the miRNA is encoded within the 3′ partner gene.

These results show that a part of the fusions reported in databases will affect miRNA genes in addition to the involved protein-coding partner genes. However, the results also suggest that a bias towards reporting in-frame fusions between protein-coding

**Table 2 MicroRNA precursors encoded in host genes with fusion transcripts in our data and in the Mitelman Database of Chromosome Aberrations and Gene Fusions in Cancer**

|  | Any fusion | 5′ Fusion partner | 3′ Fusion partner |
|---|---|---|---|
| ≥1 Tumor | 802 | 691 | 640 |
| ≥1 Tumor and in Mitelman | 320 (40%) | 192 (28%) | 162 (25%) |
| Recurrent (≥ 3 tumors) | 514 | 426 | 380 |
| Recurrent and in Mitelman | 222 (43%) | 137 (32%) | 94 (25%) |
| Recurrent in both our data and Mitelman | 9 (2%) | 4 (1%) | 4 (1%) |

Number of miRNA precursors encoded in host genes with fusion transcripts with the corresponding percentage previously reported in breast cancer in the Mitelman Database of Chromosome Aberrations and Gene Fusions in Cancer in parenthesis. 5′ or 3′ fusion partner refers to the position of the miRNA host gene and the column "any fusion" combines all precursors present in 5′ and/or 3′ fusion partner genes. Recurrent fusions were defined as occurring in at least three tumors with the host gene in the same position in our data

genes excludes a considerable part of fusions that may involve functional miRNAs.

**Multiple fusions upregulate the same miRNAs.** Our results point to a functional role in cancer for miRNAs encoded in fusion genes. To study to what extent miRNAs in fusion transcripts are aberrantly expressed, we profiled the expression of small RNAs for a subset of 186 of the original 1552 breast tumors. We then focused on cases where the expression of miRNAs in 3′ fusion partners was upregulated compared to cases without fusions involving the host gene. Due to the small number of samples available for most host gene fusions, only eight mature miRNAs reached statistical significance after correction for multiple testing, but a tentative list of 37 overexpressed miRNAs is shown in Supplementary Table 3. Some of the overexpressed miRNAs are well-known oncomiRs, e.g., *mir-21* and the *mir-106b/93/25* cluster. Almost all are involved in fusions with multiple 5′ partners (Table 3). An unusually high number of fusion transcripts is found for hsa-mir-4728, most likely related to genomic amplification of the host gene HER2[33, 34].

Gene fusions affecting *mir-33b* encoded in sterol regulatory element binding transcription factor 1 (*SREBF1*) were detected in 44 (2.8%) samples as 5′ partner and in 82 (5.3%) samples as 3′ partner (Fig. 3a). Expression of miR-33b-5p was significantly increased in all the 13 tumors with 3′ fusions of the host gene *SREBF1* for which miRNA expression data also was available ($p = 0.0001$ Student's t-test) (Fig. 3b, c). Only 1 of 13 tumors with 3′ *SREBF1* fusion and miRNA expression data was predicted to contain an in-frame fusion. Three fusion gene pairs were recurrent; one in five tumors and two in two tumors each, while the rest of the fusion pairs were singletons. In two tumor samples *SREBF1* had two different 5′ fusion partners each and three samples contained three (Supplementary Data 4).

The first intron of minichromosome maintenance complex component 7 (*MCM7*) encodes the *mir-106b/93/25* miRNA cluster and overexpression of these miRNAs has been associated with the regulation of proliferation, invasion, and migration in various human cancers[35]. Only one single case with *MCM7* gene fusion has previously been found in ovarian cancer[30]. Fourteen of our 1552 tumor samples (0.9%) had *MCM7* 3′ fusion transcripts including *mir-106b/93/25* and a single sample had a 5′ fusion (Supplementary Fig. 2). Of the six samples with *MCM7* fusions and miRNA expression data, only one had an in-frame fusion (Supplementary Data 5). One 5′ fusion partner was found in three different samples, two were found in two samples each, while the rest were singletons. In one tumor *MCM7* had fusions with two different 5′ partner genes both leading to upregulation (Supplementary Fig. 2). Overexpression of miR-25-3p was statistically significant among the six tumors with 3′ fusions ($p = 0.047$, Student's t-test). None of the other mature miRNAs reached

statistical significance in this relatively small number of samples. Examples of miRNAs with increased expression in tumors with 3′ fusion transcripts involving their host gene, but small sample numbers, are shown in Supplementary Fig. 3.

A miRNA recurrently found in fusion transcripts is *mir-21*, which is encoded in or immediately downstream of the 3′ UTR of vacuole membrane protein 1[36] (*VMP1*, also called *TMEM49*). This well-established oncogenic miRNA is upregulated in most cancers, controls several crucial cellular pathways and is a strong candidate for targeted therapy. Recurrent fusions involving *VMP1* as 3′ fusion partner have previously been reported with ribosomal protein S6 kinase B1 (*RPS6KB1*) in breast tumors[37], in the HER2-amplified breast cancer cell line BT-474[38], in esophageal adenocarcinoma[39], as well as in MCF7 breast cancer cells[40, 41]. Additional 3′ *VMP1* fusions in breast cancer include *CLTC/VMP1*[42] and *AC099850.1/VMP1*[43]. Only one of these reports described the effect of the fusion on transcription of *mir-21*[37] but the presence of the miRNA is not mentioned in either previous or subsequent papers. We detected 115 predicted fusion transcripts involving *VMP1*, 56 as 5′ partner and 59 as 3′ partner, in 32 samples. Among these, FusionCatcher classified eight as in-frame and 11 out-of-frame, while the remaining 96 consisted of other transcript types such as truncated proteins produced by the fusion of CDS to intron, UTR to CDS, etc. (Supplementary Data 6). As shown in Fig. 4a, there appears to be a concentration of 3′ fusion breakpoints closely upstream of *mir-21*, but downstream of the VMP1 protein-coding region. *VMP1* maps to 17q23.1 and fusions involving this gene have previously been associated with genomic instability on chromosome 17q[37]. We found a higher percentage of tumors with genomic amplification of the oncogene HER2 (*ERBB2*) in 17q12 among these samples (odds ratio 3.57, 95% CI [1.52, 7.96], $p = 0.0018$, Fisher's exact test; see Fig. 4b). Tumors negative for expression of estrogen receptor alpha (*ESR1*) were also overrepresented (odds ratio 2.59, 95% CI [1.03, 5.96], $p = 0.030$, Fisher's exact test). Within the TCGA data, breast cancer (odds ratio 2.26, 95% CI [1.04, 4.78], $p = 0.025$, Fisher's exact test) and lung adenocarcinoma (odds ratio 3.71, 95% CI [1.61, 8.07], $p = 0.0011$, Fisher's exact test) were more common among samples with 3′ *VMP1* fusions; breast cancer also among tumors with 5′ *VMP1* fusions (odds ratio 4.58, 95% CI [1.08, 22.14], $p = 0.019$, Fisher's exact test; see Fig. 4c).

As shown in Fig. 4d, the main mature miRNA product of the *mir-21* locus, miR-21-5p, was significantly overexpressed in tumors with 3′ *VMP1* fusion transcripts, both compared to tumors with no *VMP1* fusion transcripts and compared to tumors with only 5′ *VMP1* fusions ($p = 3.36 \times 10^{-6}$ and $p = 0.048$, respectively, Student's t-test). Tumors with only 5′ *VMP1* fusions did not have significantly higher expression of miR-21-5p than tumors without *VMP1* fusion transcripts. As shown in Fig. 4e, the expression of miR-21-3p, that is assumed to be non-functional, was also significantly higher in tumors with 3′ *VMP1* fusions,

**Table 3 MicroRNA-convergent fusions involving precursors of upregulated miRNAs**

| miRNA | Number 5′ partner genes | Number tumors with more than one 5′ partner | Number tumors 3′ fusion | Host gene | Cytogenetic location |
|---|---|---|---|---|---|
| hsa-mir-21 | 20 | 2 | 19 | VMP1 | 17q23.1 |
| hsa-mir-25, hsa-mir-93, hsa-mir-106b | 12 | 1 | 14 | MCM7 | 7q22.1 |
| hsa-mir-26a-1 | 3 | 0 | 3 | CTDSPL | 3p22.2 |
| hsa-mir-26b | 3 | 0 | 3 | CTDSP1 | 2q35 |
| hsa-mir-33b, hsa-mir-6777 | 62 | 19 | 82 | SREBF1 | 17p11.2 |
| hsa-mir-151a | 11 | 2 | 11 | PTK2 | 8q24.3 |
| hsa-mir-340 | 4 | 0 | 4 | RNF130 | 5q15 |
| hsa-mir-342 | 38 | 11 | 34 | EVL | 14q32.2 |
| hsa-mir-483 | 107 | 17 | 49 | IGF2 | 11p15.5 |
| hsa-mir-548ah, hsa-mir-4450 | 6 | 0 | 6 | SHROOM3 | 4q21.1 |
| hsa-mir-548ao | 5 | 0 | 5 | SFRP1 | 8p11.21 |
| hsa-mir-629 | 8 | 0 | 14 | TLE3 | 15q23 |
| hsa-mir-641 | 2 | 0 | 2 | AKT2 | 19q13.2 |
| hsa-mir-675 | 5 | 1 | 6 | H19 | 11p15.5 |
| hsa-mir-1229 | 4 | 1 | 3 | MGAT4B | 5q35.3 |
| hsa-mir-1249 | 6 | 2 | 29 | KIAA0930 | 22q13.31 |
| hsa-mir-1343 | 2 | 0 | 2 | PDHX | 11p13 |
| hsa-mir-3145 | 5 | 0 | 5 | NHSL1 | 6q23.3 |
| hsa-mir-3194 | 4 | 0 | 4 | NFATC2 | 20q13.2 |
| hsa-mir-3616 | 5 | 0 | 5 | EYA2 | 20q13.12 |
| hsa-mir-3909 | 3 | 0 | 3 | TOM1 | 22q12.3 |
| hsa-mir-4714 | 103 | 6 | 29 | IGF1R | 15q26.3 |
| hsa-mir-4728 | 338 | 92 | 155 | ERBB2 | 17q12 |
| hsa-mir-4802 | 5 | 0 | 5 | RBM47 | 4p14 |
| hsa-mir-6510 | 22 | 5 | 24 | KRT15 | 17q21.2 |

MicroRNA-convergent fusions involving the precursors of miRNAs upregulated in tumors with 3′ host fusions vs tumors without host gene fusions (mature miRNAs are listed in Supplementary Table 3). For clustered miRNAs, all miRNA loci included in the host gene fusions are listed

indicating transcriptional upregulation induced by the 5′ partner ($p = 4.51 \times 10^{-9}$, Student's $t$-test). The detected fusion transcripts included in- and out-of-frame fusions, as well as truncating fusions and fusions with non-coding regions such as intronic sequences or UTRs. All mir-21 overexpressing fusions were novel and singletons. Two tumors with increased expression of miR-21-5p had multiple fusion transcripts linking VMP1 to two different 5′ partner genes (Supplementary Data 6).

Most 3′ VMP1 fusions that include mir-21 lack protein-coding potential downstream of the breakpoint, which makes it possible to discriminate between the effects of the miRNA and the protein encoded by the host gene. To study the functional consequences of fusions including miR-21-5p in tumors with 3′ VMP1 fusion transcripts, we performed a GSEA[27] for miRNA target genes predicted by TargetScan 7.1[44]. Strikingly, there was a significant enrichment of predicted targets for miR-21-5p among transcripts downregulated in tumors with VMP1 fusion transcripts including mir-21 compared to tumors without mir-21 fusions with FDR < 0.001 for all targets for miR-21-5p and 0.026 for targets with at least two predicted target sites (Supplementary Fig. 4). The mRNA expression of PDCD4, an experimentally well-established target for miR-21-5p[45, 46], was also significantly lower in tumors with mir-21 fusions ($p = 0.0019$, Student's $t$-test; see Fig. 4f). Together, these results demonstrate that the inclusion of intronic miRNAs in host gene 3′ fusion transcripts can have functional consequences through deregulation of target genes.

## Discussion

We investigated the inclusion of miRNAs in breast cancer fusion genes and found these to be frequent, promiscuous in the choice of 5′ partners, and to result in overexpression. Already 13 years ago, Croce and colleagues observed that miRNA genes are frequently located in genomic regions prone to breakage in cancer[47]. Today we know that genomic rearrangements affect the expression and function of miRNAs through multiple mechanisms[48, 49]. But despite the fact that the scientific literature contains many examples of miRNAs that act as tumor drivers[8–14] and that a majority of human miRNAs are embedded in protein-coding genes, comparatively few studies have reported fusion genes leading to deregulation of intronic miRNAs[50–53]. Here we have shown that the presence of miRNAs in fusion genes is non-random and thus suggests a selective pressure associated with functional constraints. In contrast to what has previously been reported for protein-coding fusion genes we did not observe any enrichment for properties of protein-coding gene partners in miRNA-encoding fusions indicating that it is the function of the miRNA rather than the potential protein product that provides a selective advantage. There is also a preference among miRNA host genes compared to non-host genes to fuse with highly expressed 5′ partners, which would promote overexpression of the encoded miRNAs. Finally, there appears to be selection against inclusion of the miRNA when miRNA host genes are included in in-frame fusions. Since several miRNAs have been shown to regulate their own host genes through various feedback mechanisms, this could indicate that miRNAs are excluded to avoid regulation of the chimeric protein when it provides a functional advantage for the cancer cell.

The reasons for the apparent shortage of reports about the inclusion of miRNA in fusion transcripts may in part be of technical nature or related to the tools available for their investigation. However, and as shown here, we believe that at least in the context of miRNA-encoding fusions, the perception of

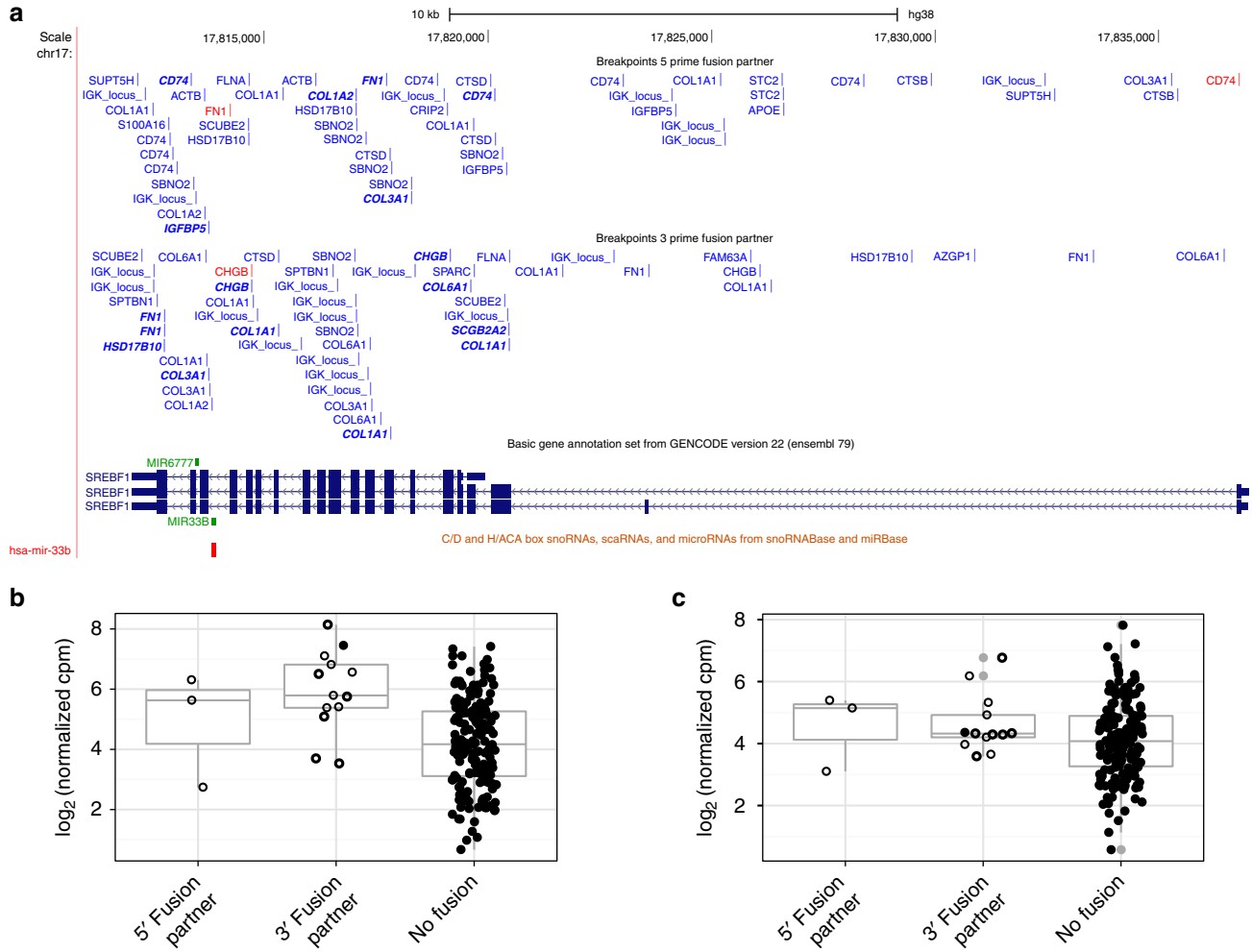

**Fig. 3** Novel fusion genes involving *mir-33b* in *SREBF1*. Breakpoints for 5′ and 3′ *SREBF1* fusion transcripts (**a**). Gene symbols refer to the corresponding partner gene for each fusion transcript. Red marks in-frame fusions between partner gene coding sequences (CDSs) and blue other fusion transcripts with out-of-frame fusions between partner gene CDSs in bold italic font. Expression of miR-33b-5p (**b**) but not miR-33b-3p (**c**) was significantly increased in 13 tumors with 3′ *SREBF1* fusions and miRNA expression data compared with tumors without fusions of the host gene. Samples with *SREBF1* fusions are shown as filled circles for in-frame fusions between partner gene CDSs, open circles with thick line for out-of-frame fusions between partner gene CDSs, and open circles with thin line for all other fusion transcripts

functionally relevant fusions has to be revised. First, most studies of fusion genes mainly focus on in-frame fusion transcripts, which excludes the majority of the fusions that could lead to deregulation of miRNA transcription. As reported for *mir-21* above, fusions leading to overexpression included in- and out-of-frame fusions, as well as truncating fusions and fusions with non-coding regions such as intronic sequences or UTRs. The eight samples with *VMP1* fusions as 3′ partner associated with over-expression of *mir-21*, however all of the 59 *VMP1* fusions that were detected have the potential to cause aberrant expression of the miRNA. Second, studies of fusion genes typically focus on the identification of recurrent chimeric transcripts with significant incidence in the analyzed samples. This is based on the assumptions that recurrence is a sign of functional significance and that selective pressure reduces diversity. Since we wished to study the impact of gene fusions on miRNA expression, we analyzed fusion transcripts irrespectively of reading frame and defined recurrence simply as the inclusion of a given miRNA in a 3′ partner gene downstream any 5′ fusion partner. The absence of any further structural constraints makes these fusions different from classical "promoter swapping" events, i.e., exchange of

regulatory control elements with preservation of the coding sequence of the 3′ partner gene. Our results do not adhere to a strict, conventional definition of recurrence. Instead, we found that different rare, conventionally non-recurrent fusions may lead to upregulation of the same miRNA. The majority of gene fusions affecting miRNAs were singletons and, if analyzed independently, their observed frequency would not be significant. By shifting the focus of analysis from coding potential to the presence of intron-encoded miRNAs, a new concept emerges that we refer to as "miRNA-convergent fusions". This is defined by the fact that structurally and functionally different 5′ fusion partners can act independently of transcript reading frame to regulate expression of the same miRNA. For example, miRNA-convergent fusions involving the *mir-21* host gene *VMP1* were found in 4.3% of all breast cancer samples analyzed at the miRNA expression level, and in more than 3% for *MCM7*, host gene of the *mir-25-106b* cluster. These findings illustrate how careful reinterpretation of apparently non-recurrent fusion events can yield potentially clinically significant results. At least for miRNA host gene fusions, recurrence should be extended beyond repeated identification of the same fusion partners to encompass fusions that include

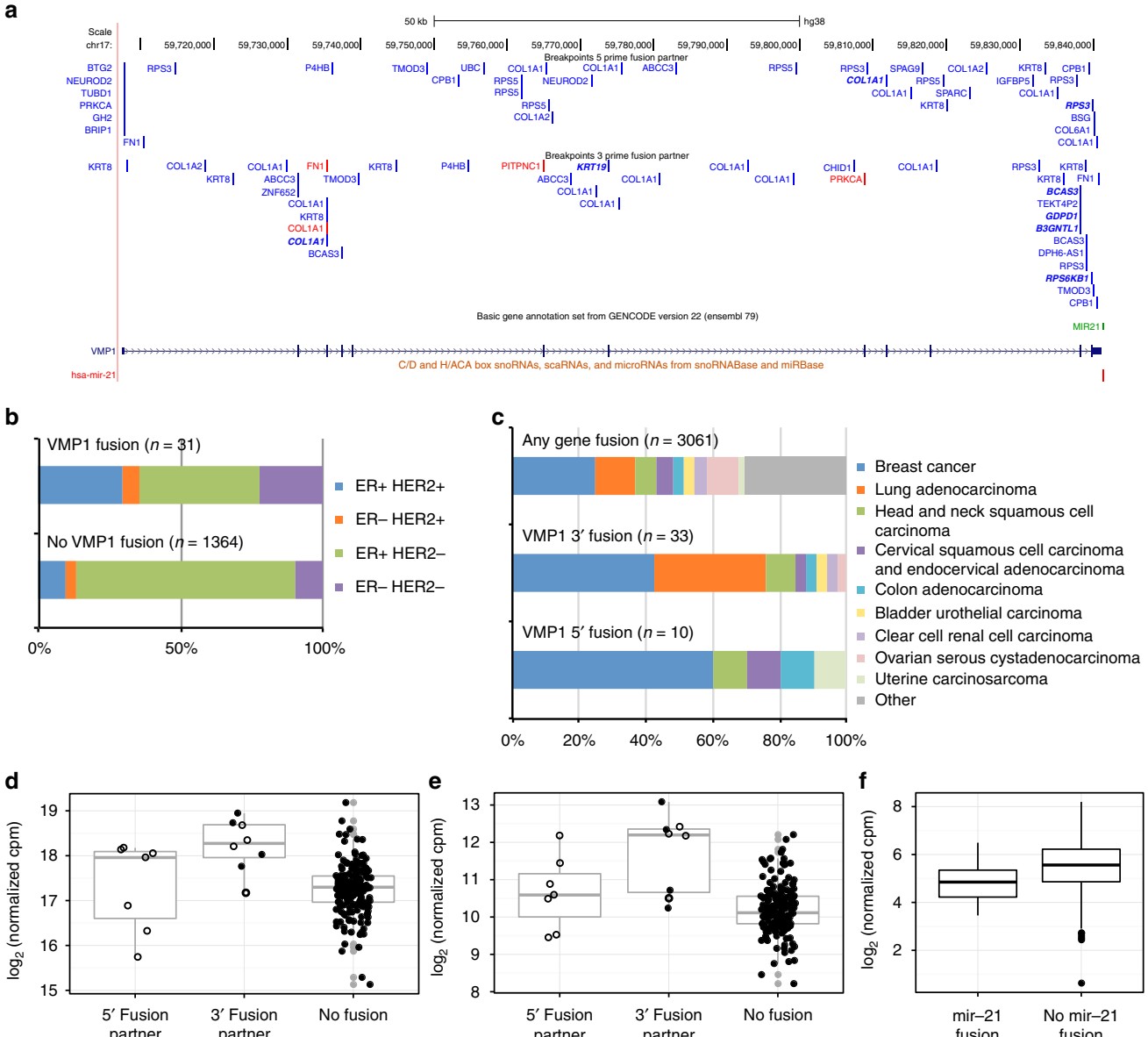

**Fig. 4** Overexpression of *mir-21* in tumors with 3′ *VMP1* fusion transcripts. Breakpoints and fusion partners for 5′ and 3′ *VMP1* fusion transcripts (**a**). Gene symbols refer to the corresponding partner gene for each fusion transcript. Red marks in-frame fusions between partner gene coding sequences (CDSs) and blue other fusion transcripts with out-of-frame fusions between partner gene CDSs in bold italic font. Tumors with genomic amplification of the oncogene *HER2* (*ERBB2*) and tumors negative for expression of estrogen receptor (ER) alpha (*ESR1*) were overrepresented among samples with *VMP1* fusion transcripts with $p = 0.0018$ and $p = 0.030$, respectively (**b**). In TCGA data, breast cancer and lung adenocarcinoma were enriched among samples with 3′ *VMP1* fusions ($p = 0.025$ and $p = 0.0011$, respectively); breast cancer also in tumors with 5′ *VMP1* fusions ($p = 0.019$, **c**). Expression of both mature miRNAs from the *mir-21* locus, miR-21-5p (**d**) and miR-21-3p (**e**), was significantly higher in tumors with 3′ *VMP1* fusions compared to tumors with no *VMP1* fusion transcripts ($p = 3.36 \times 10^{-6}$ and $p = 0.048$, respectively). Samples with *VMP1* fusions are shown as filled circles for in-frame fusions between partner gene CDSs, open circles with thick line for out-of-frame fusions between partner gene CDSs, and open circles with thin line for all other fusion transcripts. The mRNA expression of *PDCD4*, an experimentally confirmed target for miR-21-5p, was significantly lower in tumors with 3′ *VMP1* fusions including *mir-21* compared to tumors without *mir-21* fusions ($p = 0.0019$, **f**). *P*-values were calculated by Fisher's exact test for the marginal odds ratios in **b**, **c** and by Student's *t*-test in **d**-**f**

different 5′ partners but the same miRNA host gene as 3′ partner. The concept of miRNA-convergent fusions adds complexity to the heterogeneity associated with fusion genes in cancer and forces us to reconsider the classical definition of recurrence in assessment of the clinical importance of fusions.

In summary, our results add a new level of complexity to our understanding of the functional consequences of fusion genes in malignancies and in the light of these findings, we suggest that the current criteria for selection, classification, and functional evaluation

of fusion genes should be revised to also consider the impact of miRNAs.

## Methods

**Patient material**. The study was conducted in accordance with the Declaration of Helsinki and has been approved by the Regional Ethical Review Board of Lund (diary numbers 2007/155, 2009/658, 2009/659, 2014/8), the county governmental biobank center, and the Swedish Data Inspection group (diary number 364-2010). Written information was given by trained health professionals and all patients provided written informed consent.

**Strand-specific mRNA sequencing**. RNA sequencing was described[20]. Briefly, mRNA was isolated from 1.1 µg total RNA by two rounds of purification using the Dynabeads mRNA DIRECT Kit (Invitrogen/Thermo Fisher Scientific, Waltham, MA, USA) on a KingFisher Flex Magnetic Particle Processor (Thermo Fisher Scientific), Isolated mRNA was subjected to zinc-mediated fragmentation (Ambion/Thermo Fisher Scientific) by incubation in 50 µl 1× Fragmentation Reagent for 1.5 min at 70 °C immediately followed by incubation on ice and addition of 5 µl Stop Buffer to produce ~240 base pair (bp) fragments. Fragmented mRNA was purified on Zymo-Spin I-96 plates using the Oligo Clean & Concentrator Kit (Zymo Research, Irvine, CA, USA) with high EtOH (~70%) to preserve fragmented mRNA > 16 nucleotides (nt).

First strand cDNA synthesis was performed with SuperScript II reverse transcriptase using random hexamers in 20 µl reactions containing 1× First Strand buffer, 0.01 M DTT, 500 nM deoxyribonucleotide triphosphates (dNTPs), and 20 U RNaseOUT according to the manufacturer's instructions (all reagents from Invitrogen/Thermo Fisher Scientific). RNA and 1 µl random hexamers (3 µg/µl) were first denatured for 5 min at 65 °C, then chilled on ice before addition of the first strand mix and incubation for 2 min at 25 °C. After addition of 1 µl SuperScript II reactions were incubated at 25 °C for 10 min, followed by 42 °C 50 min, 70 °C 15 min and 4 °C hold. First strand clean-up was then performed with Illustra AutoScreen-96A plates (GE Healthcare, Chicago, IL, USA) to remove unincorporated dNTPs. Second strand synthesis was performed in 100 µl reactions with purified first strand cDNA (~16 µl) and 84 µl of second strand master mix containing 1.3 µl 5× First Strand buffer, 20 µl 5× Second Strand buffer, 3 µl 10 mM dNTPs with dUTP instead of dTTP, 1 µl 0.1 M DTT, 5 µl DNA Pol I (10 U/µl), 0.2 µl RNase H (10 U/µl), and 53.5 µl H$_2$O (all reagents from Invitrogen/Thermo Fisher Scientific). Reactions were incubated for 2.5 h at 16 °C before clean-up on Zymo-Spin I-96 plates with the Oligo Clean & Concentrator Kit (Zymo Research) and low EtOH (~57%) to preserve DNA > 80 bp.

The cDNA was end-repaired and A-tailed in 40 µl reactions containing 30 µl cDNA, 1× T4 ligase buffer, 500 nM dNTPs, 250 nM ATP, 5 U T4 DNA pol, 10 U T4 polynucleotide kinase, and 1 µl Taq DNA polymerase by incubation at 25 °C 20 min, 72 °C 20 min, 12 °C hold. Adapter ligation was then performed in 50 µl reactions containing 40 µl of end-repaired and A-tailed double-stranded cDNA, 1 µl of a 1:5 dilution of adapters from TruSeq DNA LT Sample Prep Kit A and B (Illumina, San Diego, CA, USA), 3 µl T4 DNA ligase (5 U/µl), 1 µl 10× T4 DNA ligase buffer, and 5 µl H$_2$O by incubation at 22 °C 30 min, 4 °C hold. Adapter-ligated cDNA was size-selected on a KingFisher Flex Magnetic Particle Processor (Thermo Fisher Scientific) using polyethylene glycol MW 8000 and carboxylic acid (CA) beads to remove fragments < 200 bp before digestion of the second cDNA strand in 15 µl reactions containing 13 µl size-selected cDNA, 1× UDG buffer, and 0.5 U uracil-DNA glycosylase. Reactions were incubated at 37 °C 15 min, 94 °C 10 min, and 4 °C hold.

Single-stranded cDNA was amplified by PCR in 45.9 µl reactions containing 15 µl UDG-digested cDNA, 2.625 µl of a 1:2 dilution of Illumina Primer Cocktail, 0.9 µl 10 mM dNTPs, 22.5 µl Phusion polymerase mix (New England Biolabs, Ipswich, MA, USA) and 4.875 µl H$_2$O using the PCR program 98 °C 3 min, 12 cycles of 98 °C 30 s, 60 °C 30 s, 72 °C 30 s, 72 °C 10 min, 4 °C hold. Fragments > 700 bp and <200 bp were excluded by two cycles of size selection on a KingFisher Flex Magnetic Particle Processor (Thermo Fisher Scientific) using polyethylene glycol MW 8000 and CA beads. Libraries were pooled and sequenced by 2 × 50 bp paired-end sequencing on a HiSeq 2000 (Illumina).

**Detection of fusion transcripts in mRNA-seq data**. FusionCatcher version 0.99.4b[19] was used to search the RNA-seq data for chimeric transcripts with the highly-sensitive option and all aligners (blat, star, bowtie2, and bwa). Fusion transcripts with any of the following descriptions were excluded from further analysis since they are associated with a high rate of false positives according to the FusionCatcher manual: banned, bodymap2, cacg, conjoing, cta_gene, ctb_gene, ctc_gene, ctd_gene, distance1000bp, ensembl_fully_overlapping, ensembl_same_strand_overlapping, gtex, hla, mt, pair_pseudo_genes, paralogs, readthrough, refseq_fully_overlapping, refseq_same_strand_overlapping, rp_gene, rp11_gene, rrna, similar_reads, similar_symbols, ucsc_fully_overlapping, and ucsc_same_strand_overlapping.

**Identification of miRNAs in fusion genes**. Fusion transcript breakpoints from the FusionCatcher analysis were combined with GENCODE[19] v22 annotation for the corresponding gene identifiers to determine 5′ and 3′ ends for both 5′ and 3′ fusion partner genes. Annotation from miRBase[21] v21 was then used to detect any pre-miRNAs fully contained within these coordinates or, if there was no identified host gene, located within 2 kb downstream of a 3′ partner gene. FusionCatcher classification was used to divide fusion transcripts into three classes "in-frame", "out-of-frame", or "other" (for all other categories). For normal samples, fusion transcript breakpoints from Babiceanu et al.[29] were converted to the hg38 genome assembly using UCSC LiftOver and combined with GENCODE v22 annotation to determine the ends of partner genes. For TCGA samples, fusion transcript breakpoints from were converted to the hg38 genome assembly using UCSC LiftOver and combined with Entrez annotation to determine the ends of partner genes. Since no breakpoint coordinates were available for the Mitelman Database of Chromosome Aberrations and Gene Fusions in Cancer[30], miRNA host genes in fusions were identified through matching by gene symbol. For FusionCancer[28] the analysis was limited to genes with ENSEMBL gene identifiers and fusion support classes A, B, and C. Breakpoint coordinates were converted to the hg38 genome assembly using UCSC LiftOver[54] and combined with GENCODE v22 annotation. For fusions in the COSMIC database[32], genomic coordinates were calculated from the inferred mRNA breakpoints for fusion transcripts with ENSEMBL transcript identifiers using GENCODE v22 annotation. Fusions were excluded if the genomic coordinates could not be calculated due to inconsistencies between COSMIC mRNA positions and GENCODE transcript annotation.

**Gene set enrichment analysis**. For GSEA[27] of fusion transcripts, Bioconductor tools (AnnotionDbi, org.Hs.eg.db) were used for mapping ENSEMBL to Entrez identifiers for the GENCODE v22 gene annotation files. The clusterProfiler[28], DOSE and ReactomePA Bioconductor packages were used and Benjamini–Hochberg correction was used to control the false discovery rate. MSigDB v5.1 files were downloaded from Broad Institute (http://software.broadinstitute.org/gsea/msigdb). For GSEA of predicted miR-21-5p targets from TargetScan 7.1[44]using the javaGSEA desktop application (http://software.broadinstitute.org/gsea/), a ranked table of log$_2$ fold changes was calculated for all GENCODE transcripts that were included in the TargetScan prediction as representative transcripts and had a median expression across all 1552 tumors of ≥1 FPKM (fragments per kilobase million).

**Small RNA sequencing**. Total RNA was extracted from aliquots of the tumor homogenates used for mRNA-sequencing[20] using TRIzol LS (Thermo Fisher Scientific) according to the manufacturer's instructions and concentrations were measured on a NanoDrop ND-1000 spectrophotometer (NanoDrop Technologies Inc., Wilmington, DE, USA). For small RNA sequencing, 500 ng total RNA and 20 pmol of pre-adenylated 3′ adapter were denatured for 2 min at 70 °C before ligation for 1 h at 25 °C in an 18 µl reaction containing 200 U T4 RNA ligase 2, truncated K227Q, 1× buffer for T4 RNA ligase, 15% PEG MW 8000 (New England Biolabs), and 40 U RiboLock RNase inhibitor (Thermo Fisher Scientific). Next, 20 pmol of reverse transcription primer was added and annealed by incubation for 5 min at 75 °C, 15 min 37 °C, and 15 min 25 °C. For each reaction, 20 pmol of 5′ adapter was denatured separately for 2 min at 70 °C and then ligated in 25 µl reactions in the presence of 10 U T4 RNA ligase 1, 1× buffer for T4 RNA ligase, and 1 mM ATP (New England Biolabs) for 1 h at 25 °C. Reverse transcription was performed in 40 µl reactions with 200 U ProtoScript II reverse transcriptase in 1× reaction buffer, 10 mM DTT, and 500 µM dNTPs (New England Biolabs). In total, 20 µl of cDNA was used for PCR amplification in 50 µl reactions with 2× LongAmp Taq Master Mix (New England Biolabs) and 250 nM each of indexed forward (8 nt index; total size 70 nt) and reverse primers (6 nt index; total size 63 nt). Pooled libraries were sequenced on a NextSeq 500 with High Output v2 75 cycle-kits (Illumina). Sequences were demultiplexed using Picard and aligned against hg38 using Novoalign with settings -a TGGAATTCTCGGGTGCCAAGG -l 14 -h -1 -1 -t 90 -g 50 -x 15 -o SAM -o FullNW -r All 51 -e 51. miRNA expression was analyzed using custom Perl scripts and calculated as counts per million miRNA reads (cpm).

**Statistical analysis**. Fisher's exact test was used to calculate the significance of the overlap between sets of miRNAs and the overrepresentation of classes such as HER2-positive tumors among samples with *VMP1* fusion transcripts. Logistic regression to model the log odds of a gene being involved in fusion events was performed with the two models fusion ~ miRNA host gene+log$_2$(size) and fusion ~ miRNA host gene+log$_2$(size)+miRNA host gene:log$_2$(size) for protein-coding genes expressed in the mRNA-sequencing data. For comparison, RepeatMasker annotation was obtained from the UCSC Table Browser and analyzed using custom Perl scripts to identify genetic elements that occur within protein-coding genes at a rate similar to that of miRNAs. Student's *t*-test was used to test for differences in the expression of miRNAs between groups of tumors after log-transformation of the cpm values. Bonferroni correction was used to adjust p-values for multiple testing. For the GSEA analysis, Benjamini–Hochberg correction was used to control the false discovery rate. Statistical analyses were performed in R.

**Data availability**. The data required to replicate the findings of this study are available from NCBI Gene Expression Omnibus (GEO) under accession number GSE100769 and from the corresponding author on request.

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

## Acknowledgements

This work was supported by grants from the Berta Kamprad Foundation and the Swedish Cancer Society.

## Author contributions

H.P. and A.P. performed the small RNA sequencing. H.P. designed and performed the bioinformatic analysis. R.S. contributed to the analysis. J.H. performed the identification of fusion transcripts and together with J.V.-C. and Å.B. provided the RNA-seq data. A.K. contributed to the statistical analysis. F.M. (Mertens) and F.M. (Mitelman) participated

in the design of the study. M.H. contributed to the design of the study. C.R. conceived the idea, designed and coordinated the study. H.P., M.H., and C.R. drafted the manuscript and all authors provided comments on the final version.

## Additional information

**Competing interests:** The authors declare no competing financial interests.

