## [Peer Review File · Nature Communications]

Reviewers' Comments:

Reviewer #1:

Remarks to the Author:

The manuscript by Persson and colleagues is a very important one, as discovered the missing link in the genetics of miRNA dysregulation in cancer – although over a decade ago found to be located in cancer associated genomic regions, miRNAs, either as precursor genes or as primary transcripts, were rarely reported to be members of fusion events. The authors presented strong evidence for the frequent involvement of miRNA-host genes in fusion gene events that do not affect the protein coding gene, but the expression of the host microRNA. This is a very significant advance in the field of miRNA genomics, from a group with seminal contribution to the understanding of chromosome alterations in cancer and consequent fusion events.

The paper has multiple strengths:

- 1 the large amount of samples analyzed, all from breast cancer
- 2 the authors used deep sequencing data analyzed with adequate tools for the identification of fusions
- 3 the use of an independent confirmation in the TCGA dataset
- 4 the identification of active miRNA overexpression in the cases with fusion
- 5 the inclusion in the discussions of a paragraph explaining the possible causes why such a study was not performed till present time
- 6 the clearly written science that can be understood also by the scientists not used with this particular topics, as the journal has readers with many expertise.

Therefore, this is a manuscript of great interest for the large spectrum of readers of the journal!

There are few minor points:

- 1 what are the functions of the host genes harboring miRNAs involved in fusions, is enrichment in some Gene Ontology terms?
- 2 did the tumors with miRNA host genes fusions present some clinical peculiarities as a general class (as for each specific miRNA the number of tumors is quite small for clinical statistics)?
- 3 there are common targets for multiple miRNAs affected by fusions that can explain the involvement in breast cancer? Lets say, groups of 4 or 5 such miRNAs to have an important set of targets such as PDCD4, etc.

Reviewer #2:

Remarks to the Author:

Persson et al perform an interesting analysis of breast cancer RNAseq data with a focus on fusion transcripts involving miRNAs.

They demonstrate that miRNA host genes are more likely to participate in fusion transcripts than non-host genes. They also hypothesized that because miRNAs may regulate their target genes, that they would be less frequent within the 3' partner genes of in-frame fusions, thus reducing potential down-regulation of the aberrant fusion transcript, which they confirm with statistics. In support of this hypothesis, for in-frame fusions in which there was a miRNA within the 3' partner, the expression level of the 5' partner was typically greater, potentially compensating for down-regulation by the miRNA.

They further go on to show that some miRNAs are recurrently expressed in a diversity of fusion transcripts and that this results in overexpression of the miRNA sequence in small RNAs, indicating that the processed miRNAs are also overexpressed. This data, while interesting was not compelling

as to a functional role for these miRNAs affected by convergent fusions as they either affected a small number of total breast cancer tumors or were present within genes with proposed protein functions relevant to cancer (i.e. VMP1 or ERBB2).

The work would be strengthened considerably by experimental confirmation of functional significance of the mir-21 overexpression in tumors in which there is a fusion transcript with mir-21 as distinct from effects on the protein coding genes in a subset of tumors.

Minor comments:

1. There appears to be a discrepancy between the fraction of host genes identified in fusions transcripts in the authors' cohort and the TCGA. Can the authors provide a rationale?
2. Figure 3. Part a is difficult to interpret and the legend is confusing.
3. Figure 4. Some statistics would improve parts b and c.
4. The authors do not describe their methods for multiple hypothesis correction and only mention it in the context of estimating the statistical significance of miRNA overexpression but multiple hypothesis correction is needed in multiple portions of the manuscript. Perhaps adding more detail in the methods would suffice to address this issue.

Reviewer #3:

Remarks to the Author:

Major Issues:

1. Regarding the intersections with TCGA (Page 5), the authors were using the curated list provided by TCGA for pan-cancer, however, they should show the intersection with TCGA's BRCA data only. In addition to further strengthening the authors' point about the validity of miRNA fusion targets, this analysis will provide another independent assessment of the consistency of fusion calling across different institutions. This is a non-trivial point because RNA quality and library preparation can certainly introduce artifacts masquerading as fusions. If this analysis were done, would the overlap still be significant?
2. The authors need to provide a citation for there statement "However, since longer genes are also more likely to encompass miRNAs...". If this is true, would the density of miRNAs be consistent throughout the whole gene, or there would there be any bias toward a specific region, for example, the first few exons?
3. Are the 3' fusion genes harboring miRNA significantly longer than the non-harboring-miRNA 3'fusion genes?
4. The authors state "Enrichment of miRNA host genes in fusions is not associated with protein functions". Even though this is a negative result, the authors should still show in the supplement the p-values of top 5 terms.
5. Regarding fusion calling, how much does the amount of fusions vary when the detection thresholds varies?
6. How conserved are the fusion-targeted miRNAs compared to the rest?

Reviewer #1 Expert in miRNA and cancer:

The manuscript by Persson and colleagues is a very important one, as discovered the missing link in the genetics of miRNA dysregulation in cancer – although over a decade ago found to be located in cancer associated genomic regions, miRNAs, either as precursor genes or as primary transcripts, were rarely reported to be members of fusion events. The authors presented strong evidence for the frequent involvement of miRNA-host genes in fusion gene events that do not affect the protein coding gene, but the expression of the host microRNA. This is a very significant advance in the field of miRNA genomics, from a group with seminal contribution to the understanding of chromosomal alterations in cancer and consequent fusion events.

The paper has multiple strengths:

- 1 the large amount of samples analyzed, all from breast cancer
- 2 the authors used deep sequencing data analyzed with adequate tools for the identification of fusions
- 3 the use of an independent confirmation in the TCGA dataset
- 4 the identification of active miRNA overexpression in the cases with fusion
- 5 the inclusion in the discussions of a paragraph explaining the possible causes why such a study was not performed till present time
- 6 the clearly written science that can be understood also by the scientists not used with this particular topics, as the journal has readers with many expertise.

Therefore, this is a manuscript of great interest for the large spectrum of readers of the journal!

There are few minor points:

1. what are the functions of the host genes harboring miRNAs involved in fusions, is enrichment in some Gene Ontology terms?

Response:

As noted by the reviewer the observed enrichment of miRNA hosts in fusions may be due to the function of protein encoded by the host rather than for the presence of a miRNA. However, we analyzed this point and found no significant or in some case very marginal enrichment of the proteins encoded by the miRNA hosts for Gene Ontology (GO) terms (cellular component, molecular function, and biological process), Disease Ontology terms (DOSE), the Kyoto Encyclopedia of Genes and Genomes (KEGG), the Reactome Pathway Database, or the Molecular Signatures Database (MSigDB). The results are reported in the text and the corresponding p-values of top 5 terms can be found in the Supplemental Table 5.

2. did the tumors with miRNA host genes fusions present some clinical peculiarities as a general class (as for each specific miRNA the number of tumors is quite small for clinical statistics)?

Response:

We looked for association with clinical characteristics such as ER, PR, Her2 status and grade.

We cannot see a trend if we analyze all convergent fusions together. However, it seems as there is an association between miR-21 convergent fusions and Her2 status and high grade (see results below). However, as noticed by the reviewer the number of mir-21 fusions is quite small to come to certain conclusions.

```
> #####  
## Her2-status ##  
#####  
HER2.table.miR21  
      HER2  
has.mi21 Amplifiering, terapiindikation Ej amplifierat  
FALSE      187      1200  
TRUE        8        10
```

```
> fisher.test(HER2.table.miR21)  
Fisher's Exact Test for Count Data  
data: HER2.table.miR21  
p-value = 0.001508  
alternative hypothesis: true odds ratio is not equal to 1  
95 percent confidence interval:  
0.06833481 0.57682533  
sample estimates:  
odds ratio  
0.1951511
```

```
#####  
## Grade 1-status ##  
#####  
> NHG1.table.miR21
```

```
      NHG1  
has.mi21 FALSE TRUE  
FALSE 1269 243  
TRUE   19  0
```

```
> fisher.test(NHG1.table.miR21)  
  
      Fisher's Exact Test for Count Data  
  
data: NHG1.table.miR21  
p-value = 0.05787  
alternative hypothesis: true odds ratio is not equal to 1  
95 percent confidence interval:  
0.000000 1.128645  
sample estimates:  
odds ratio  
0
```

```
#####  
## Grade 2-status ##  
#####
```

```
> NHG2.table.miR21
```

```
      NHG2  
has.mi21 FALSE TRUE  
FALSE  804  708  
TRUE   14   5
```

```
> fisher.test(NHG2.table.miR21)
```

Fisher's Exact Test for Count Data

```
data: NHG2.table.miR21  
p-value = 0.1037  
alternative hypothesis: true odds ratio is not equal to 1  
95 percent confidence interval:  
0.1138119 1.1997402  
sample estimates:  
odds ratio  
0.4058061
```

```
#####  
## Grade 3-status ##  
#####
```

```
> NHG3.table.miR21
```

```
      NHG3  
has.mi21 FALSE TRUE  
FALSE  951  561  
TRUE   5   14
```

```
> fisher.test(NHG3.table.miR21)
```

Fisher's Exact Test for Count Data

```
data: NHG3.table.miR21  
p-value = 0.001543  
alternative hypothesis: true odds ratio is not equal to 1  
95 percent confidence interval:  
1.603113 16.912970  
sample estimates:  
odds ratio  
4.741626
```

3. there are common targets for multiple miRNAs affected by fusions that can explain the involvement in breast cancer? Lets say, groups of 4 or 5 such miRNAs to have an important set of targets such as PDCD4, etc.

Response:

This is an interesting point.

We addressed the question but it is difficult to draw a solid conclusion from the results. We searched a functional overlap between the 26 miRNAs overexpressed by a fusion using validated targets filtered through literature curation (miRecords and TarBase). The predicted targets of half of these miRNAs (miR-151a, miR-197-3p, miR-21-5p, miR-25-3p, miR-33b-5p, miR-340-5p, miR-342, miR-33b-5p, miR-340-5p, miR-342, miR-592, miR-641 and miR-93-5p) were enriched in genes involved in MAPK signaling. But as we stated in the text we also have a relatively high percentage of tumors with genomic amplification of HER2/ERBB2. This sampling bias makes difficult the functional interpretation of the fusions. I suppose we need more data before we can see clear trends, if any.

Reviewer #2 Expert in fusion genes:

Persson et al perform an interesting analysis of breast cancer RNAseq data with a focus on fusion transcripts involving miRNAs.

They demonstrate that miRNA host genes are more likely to be participate in fusion transcripts than non-host genes. They also hypothesized that because miRNAs may regulate their target genes, that they would be less frequent within the 3' partner genes of in-frame fusions, thus reducing potential down-regulation of the aberrant fusion transcript, which they confirm with statistics. In support of this hypothesis, for in-frame fusions in which there was a miRNA within the 3' partner, the expression level of the 5' partner was typically greater, potentially compensating for down-regulation by the miRNA.

They further go on to show that some miRNAs are recurrently expressed in a diversity of fusion transcripts and that this results in overexpression of the miRNA sequence in small RNAs, indicating that the processed miRNAs are also overexpressed. This data, while interesting was not compelling as to a functional role for these miRNAs affected by convergent fusions as they either affected a small number of total breast cancer tumors or were present within genes with proposed protein functions relevant to cancer (i.e. VMP1 or ERBB2).

The work would be strengthened considerably by experimental confirmation of functional significance of the mir-21 overexpression in tumors in which there is a fusion transcript with mir-21 as distinct from effects on the protein coding genes in a subset of tumors.

Response:

The confirmation of a functional role for miR-21 convergent fusions strengthened indeed our findings. We thank Reviewer #2 for bringing out the point.

Following the Reviewer's suggestion we analyzed the functional role of miR-21 convergent fusions. Here, the protein coding capacity of the VMP1 part gene involved in the fusion is mainly destroyed so that we can discriminate between the function of the miRNA or the protein encoded by the host.

Bartel and coworkers (Nature, 2010) demonstrated that miRNA predominately act by regulating the mRNA level of their targets. We therefore used the RNA-seq data to analyze expression differences of

predicted miR-21 target genes in fusions including mir-21 3' of the breakpoint compared to tumors without mir-21 fusions. We observed that, as expected, there was a significant enrichment of predicted targets for miR-21-5p among transcripts downregulated in tumors with miR-21 convergent fusions.

PDCD4 is one of the best-characterized targets of miR-21. Therefore, we also measured the expression levels of PDCD4 in relation of miR-21 fusions. Here again, there is a significant lower level of PDCD4 in samples with a miR-21 fusion transcript compared to samples without a fusion.

We believe these results support our general conclusions about a functional role for miRNA-convergent fusions. The results are now summarized in the main text and graphically displayed in a new figure 4f and Supplementary Fig 4.

Minor comments:

1. There appears to be a discrepancy between the fraction of host genes identified in fusions transcripts in the authors' cohort and the TCGA. Can the authors provide a rationale?

Response:

As noted by the reviewer, the fraction of miRNA host genes identified in fusions was higher in our data compared with the TCGA data; 61 vs 32% for all fusions and 39 vs 6% for fusions recurrent in at least three samples. However, the total number of fusion events reported by Yoshihara et al. (Oncogene 2015) for the TCGA samples was also considerably smaller. The average number of genes identified as fusion partners per tumour was 81 for our dataset and 9 among breast tumours in the TCGA data (7 for all cancer types). These differences can be explained by the very strict criteria that were used by Yoshihara et al. to further filter the predicted fusion transcripts, e.g. transcript allele frequency, partner gene variety, and the presence of nearby breakpoints in DNA copy number data.

Furthermore, fusion transcripts were reported using Entrez gene identifiers in the Yoshihara et al. Supplementary Table 1, while we have used the more comprehensive GENCODE annotation. Using Entrez annotation, 1108 out of 1881 miRNA precursor loci could be annotated with an overlapping host gene, compared to 1320 using GENCODE v22 coordinates. The total number of miRNAs identified in fusion transcripts is therefore also lower in the TCGA data.

2. Figure 3. Part a is difficult to interpret and the legend is confusing.

Response:

As suggested by the reviewer, the correction has been made. The legend for Figure 3a has been changed to "Breakpoints for 5' and 3' SREBF1 fusion transcripts (a). Gene symbols refer to the corresponding partner gene for each fusion transcript. Red marks in-frame fusions between partner gene coding sequences (CDSs) and blue other fusion transcripts with out-of-frame fusions between partner gene CDSs in bold italic font." The same additional information has been included in the legend for Figure 4a. We hope this will make the figures easier to interpret.

We have also rephrased the legend for Figure 2a slightly for increased clarity. It now reads “Cumulative distribution of the fraction of in-frame fusions among all fusion transcripts, plotted separately for 5' and 3' partner genes that lack or include miRNAs within the fused gene segments in our breast cancer data”.

3. Figure 4. Some statistics would improve parts b and c.

Response:

Since Figure 4b shows samples divided by both HER2- and ER-status, we have now also included a Fisher test for overrepresentation of estrogen receptor-negative tumors (ER-). The Results section referring to Figure 4b now reads: “We found a higher percentage of tumors with genomic amplification of the oncogene HER2 (ERBB2) in 17q12 among these samples (odds ratio 3.57, 95% CI [1.52, 7.96], $p = 0.0018$; see Figure 4b). Tumors negative for expression of estrogen receptor alpha (ESR1) were also overrepresented (odds ratio 2.59, 95% CI [1.03, 5.96], $p = 0.030$).” The legend for Figure 4b has been modified to “Tumors with genomic amplification of the oncogene HER2 (ERBB2) and tumors negative for expression of estrogen receptor (ER) alpha (ESR1) were overrepresented among samples with VMP1 fusion transcripts with $p = 0.0018$ and $p = 0.030$, respectively (**b**)”. To avoid repetition the sentence “P-values were calculated by Fisher’s exact test for the marginal odds ratios in b, c and by Student’s t-test in d, e.” has been added to the end of the legend for Figure 4.

For Figure 4c we now test for overrepresentation among samples with VMP1 fusions for breast (5' and 3') and lung cancer (3'). The remaining categories were not tested due to the very low number of fusion events. This analysis has been added to the Results section as “Within the TCGA data, breast cancer (odds ratio 2.26, 95% CI [1.04, 4.78], $p = 0.025$) and lung adenocarcinoma (odds ratio 3.71, 95% CI [1.61, 8.07], $p = 0.0011$) were more common among samples with 3' VMP1 fusions; breast cancer also among tumors with 5' VMP1 fusions (odds ratio 4.58, 95% CI [1.08, 22.14], $p = 0.019$; see Figure 4c). Marginal odds ratios were calculated and tested using Fisher’s exact test.” The legend has been changed to “In TCGA data, breast cancer and lung adenocarcinoma were enriched among samples with 3' VMP1 fusions ($p = 0.025$ and $p = 0.0011$, respectively); breast cancer also in tumors with 5' VMP1 fusions ($p = 0.019$, **c**).”

We hope that these additions help to clarify the results presented in Figure 4b-c.

4. The authors do not describe their methods for multiple hypothesis correction and only mention it in the context of estimating the statistical significance of miRNA overexpression but multiple hypothesis correction is needed in multiple portions of the manuscript. Perhaps adding more detail in the methods would suffice to address this issue.

Response:

The correction has been made. We have now included the missing information on multiple hypothesis correction in the paragraphs on GSEA and statistical analysis in the Methods section.

Reviewer #3 Expert in cancer genomics:

Major Issues:

1. Regarding the intersections with TCGA (Page 5), the authors were using the curated list provided by TCGA for pan-cancer, however, they should show the intersection with TCGA's BRCA data only. In addition to further strengthening the authors' point about the validity of miRNA fusion targets, this analysis will provide another independent assessment of the consistency of fusion calling across different institutions. This is a non-trivial point because RNA quality and library preparation can certainly introduce artifacts masquerading as fusions. If this analysis were done, would the overlap still be significant?

Response:

We have now added a comparison with only TCGA breast cancer data to the manuscript. The overlap with our data remains significant and the corresponding tests have been added to the Results section. Table 1 has furthermore been updated with additional information on the number of analysed miRNA precursors with supporting TCGA breast tumour fusions. While we fully agree that the TCGA breast cancer data is the most relevant material for validation, we also believe that the comparison with all cancer types can contribute important information for specific miRNAs. One example is the high frequency of VMP1/mir-21 fusions in lung adenocarcinoma in addition to breast cancer. For reasons described in the response to reviewer #2, minor comment 1, the number of reported fusion events is also considerably smaller in the TCGA data. This can to some extent be compensated by comparing our results to the larger set of all cancer samples.

2. The authors need to provide a citation for their statement "However, since longer genes are also more likely to encompass miRNAs...". If this is true, would the density of miRNAs be consistent throughout the whole gene, or there would there be any bias toward a specific region, for example, the first few exons?

Response:

The correction has been made. In the revised version we cite the works by Hinske et al 2010 and Biasiolo et al 2011. In both papers they show that miRNA hosts are significantly longer than non-hosts. We controlled their conclusions by doing our own calculations and also found that protein coding genes harboring miRNA are indeed, on average, longer than genes not harboring miRNA (non-harboring-miRNA median: 21400 bp vs harbouring-mirna median: 87055 bp, Wilcoxon test: $p < 2.2 \times 10^{-16}$).

In response to the second point, Hinske et al observed that 65.5% of host genes had miRNAs in the first five introns. However, it seems as this is more a trend rather than a bias ($p = 0.030$ in chi-squared test as reported in their paper).

3. Are the 3' fusion genes harboring miRNA significantly longer than the non-harboring-miRNA 3'fusion genes?

Response:

Yes, the size difference between host and non-host genes is significant for all 3' fusion partner genes (\log_2 fold change 0.82, $p < 2.20 \times 10^{-16}$, Student's t-test) and for recurrent 3' fusion partners (\log_2 fold change 0.78, $p = 3.67 \times 10^{-16}$, Student's t-test). A new figure has been included in the manuscript as Supplementary Figure 1 to show the size distributions for genes divided by fusion and miRNA host gene status.

4. The authors state "Enrichment of miRNA host genes in fusions is not associated with protein functions". Even though this is a negative result, the authors should still show in the supplement the p-values of top 5 terms.

Response:

The p-values for the top 5 terms are showed in Supplementary Table 5.

5. Regarding fusion calling, how much does the amount of fusions vary when the detection thresholds varies?

Response:

Current versions of FusionCatcher automatically selects sensitivity parameters and does not allow users (trivially) to vary the fusion detection sensitivity. We installed a previous version (0.99.4b) that enables the user to try different fusion detection sensitivity settings.

In the table below we list counts for four different fusion detection settings - settings; 1 "default", 2 "sensitive", 3 "mildly sensitive", 4 "highly sensitive", more false positive candidates with higher parameter setting number. The fusion analysis in our paper uses the "highly sensitivity" parameter setting.

	All samples				miR-21			
	1	2	3	4	1	2	3	4
Mean	8	29	113	218	30	128	483	900
Median	2	4	16	33	10	24	78	133
Stdev	24	95	397	731	48	189	826	1520

As expected the fusion candidate count increases as the sensitivity for fusion detection is increased. The average and median number of fusions are much larger for miR-21 samples compared to the overall data set, and the average/median ratio (miR/all) is around 5 for all parameter settings.

6. How conserved are the fusion-targeted miRNAs compared to the rest?

Response:

This is an interesting point although difficult to evaluate from a functional perspective. The accepted interpretation is that high evolutionary conservation is a sign of functional importance because they are the product of purifying selection. However, non-conserved miRNAs may also control important gene regulatory mechanisms. This has been shown in plants but also in humans. For instance, we have shown that even non-conserved, primate specific miRNAs may be responsible of important oncogenic functions (Newie et al Sci Rep 2016). Lund AH and coworkers showed that miR-95, although only found in primates, could regulate highly conserved genes such as SUMF1 blocking autophagy-mediated degradation (Frankel et al Nat. Commun. 2014). Even species-specific miRNAs are functionally important in neurogenesis (Mor E et al 2011. Nucleic Acids Res), apoptosis (Tang R et al. 2012. Cell Res 22: 504–15) or pluripotency (Judson RL et al. 2009. Nat Biotechnol).

We calculated the degree of conservation using PhastCons and PhyloP scores based on the alignment of 20 mammalian genomes for the precursor miRNAs. We compared conservation of miRNA precursors involved in fusions as 5' and 3' partner with miRNAs not involved in any fusion. MiRNAs in the 5' partner and in the 3' partner display a similar evolutionary behavior to those miRNAs not involved in fusions. There is perhaps a little higher degree of conservation for miRNA encoded in the 5' partner. Although significant, the difference is marginal so we decided not to include these results in the revised version.

We want to thank the Editor and the reviewers for the time spent on improving the quality of our manuscript.

Carlos Rovira
Lund University
Oncology/Pathology

Reviewers' Comments:

Reviewer #1:

Remarks to the Author:

The authors responded adequately to the comments and the paper is of great interest for the readers of the journal. This is carefully done research!

Reviewer #2:

Remarks to the Author:

The authors have appropriately responded to my comments and those of the other reviewers. This manuscript is ready for acceptance in my opinion and contributes to our understanding of the impact of structural variation on miRNA expression.

Reviewer #3:

Remarks to the Author:

The authors have addressed all reviewer concerns in the revised version of the manuscript.